# Prior physical illness predicts death better than acute physiological derangement on intensive care unit admission in COVID-19: A Swedish registry study

Karl Stattin[1]*, Michael Hultström[1,2,3,4], Robert Frithiof[1], Miklos Lipcsey[1,5],
Rafael Kawati[1]

1 Department of Surgical Sciences, Anaesthesiology and Intensive Care, Uppsala University, Uppsala, Sweden, 2 Department of Medical Cell Biology, Integrative Physiology, Uppsala University, Uppsala, Sweden, 3 Department of Epidemiology, Biostatistics and Occupational Health, McGill University, Montréal, Québec, Canada, 4 Lady Davis Institute of Medical Research, Jewish General Hospital, McGill University, Montréal, Québec, Canada, 5 Hedenstierna Laboratory, Department of Surgical Sciences, Uppsala University, Uppsala, Sweden

* karl.stattin@surgsci.uu.se

**Data Availability Statement:** The entire dataset is available to researchers upon request with attached

## Abstract

COVID-19 is associated with prolonged intensive care unit (ICU) stay and considerable mortality. The onset of persistent critical illness, defined as when prior illness predicts death better than acute physiological derangement, has not been studied in COVID-19. This national cohort study based on the Swedish Intensive Care Registry (SIR) included all patients admitted to a Swedish ICU due to COVID-19 from 6 March 2020 to 9 November 2021. Simplified Acute Physiology Score-3 (SAPS3) Box 1 was used as a measure of prior illness and Box 3 as a measure of acute derangement to evaluate the onset and importance of persistent critical illness in COVID-19. To compare predictive capacity, the area under receiver operating characteristic (AUC) of SAPS3 and its constituent Box 1 and 3 was calculated for 30-day mortality. In 7 969 patients, of which 1 878 (23.6%) died within 30 days of ICU admission, the complete SAPS3 score had acceptable discrimination: AUC 0.75 (95% CI 0.74 to 0.76) but showed under prediction in low-risk patients and over prediction in high-risk patients. SAPS3 Box 1 showed markedly better discrimination than Box 3 (AUC 0.74 vs 0.65, $P<0,0001$). Using custom logistic models, the difference in predictive performance of prior and acute illness was validated, AUC 0.76 vs AUC 0.69, p<0.0001. Prior physical illness predicts death in COVID-19 better than acute physiological derangement during ICU stay, and the whole SAPS3 score is not significantly better than just prior illness. The results suggests that COVID-19 may exhibit similarities to persistent critical illness immediately from ICU admission, potentially because of long median ICU length-of-stay. Alternatively, the variables in the acute physiological derangement model may not adequately capture the severity of illness in COVID-19.

ethical approval from the Swedish Intensive Care Registry, at www.icuregswe.org. Swedish law does not support sharing of individual level registry data, and thus our ethical approval from the Swedish Ethical Review Authority does not support public uploading of individual-level data as it contains sensitive patient information.

**Funding:** The study was funded by the SciLifeLab/ Knut and Alice Wallenberg national COVID-19 research program (M.H.: KAW 2020.0182, KAW 2020.0241, https://kaw.wallenberg.org/), the Swedish Heart-Lung Foundation (M.H.: 20210089, 20190639, 20190637, www.hjart-lungfonden.se/), the Swedish Research Council (R.F.: 2014-02569, 2014-07606, www.vr.se), the Swedish Society of Medicine (M.H.: SLS-938101, www.sls.se) and the Swedish Kidney Foundation (R.F. F2020-0054, www.njurfonden.se). The funders had no role in study design, data collection and analysis, decision to publish, or preparation of the manuscript.

**Competing interests:** The authors have declared that no competing interests exist.

# Introduction

The COVID-19 pandemic led to unprecedented strain on intensive care unit (ICU) resources, partly due to the high number of patients but also due to the prolonged ICU stay and poor prognosis among COVID-19 patients [1–5]. In non-COVID-19 populations, a proportion of ICU patients has been shown to develop a condition of persistent critical illness associated with prolonged ICU stay and poor prognosis [6] but it is not known to what extent persistent critical illness contributes to the long ICU length of stay and high mortality in COVID-19. Persistent critical illness has been defined as when prior comorbidities predict death better than the severity of the acute illness, which in a variety of non-COVID-19 cohorts occurs after 5–11 days [7–11]. Previous studies in COVID-19 have found that 46–55% of patients develop persistent critical illness [12, 13], compared to 5–34% in other diagnoses [7–11], but have arbitrarily defined persistent critical illness as after 10 [12] or 21 [13] days of ICU stay.

The illness severity score Simplified Acute Physiology Score 3 (SAPS3) is widely used to predict mortality in ICU patients. SAPS3 divides predictive information into three "Boxes": 1: Previous health status; 2: Circumstances of ICU admission; and 3: Acute physiological derangement [14]. The few large validation studies performed in COVID-19 have indicated acceptable discrimination, i.e. area under receiver operating characteristic curve (AUC) 0.69 to 0.84 [15–17], but have not compared discrimination across SAPS3 Boxes or investigated if SAPS3 equally well predicted death across all waves of COVID-19. Facing the prospect of future waves of COVID-19 or other viral diseases causing acute respiratory failure, it is of interest to investigate if SAPS3 predicts death equally well across previous waves of COVID-19, with their differing patient case mixes and viral strains. Furthermore, if the SAPS3 Boxes display acceptable discrimination, they could be used to find the onset of persistent critical illness in COVID-19 using a well-known risk prediction score.

The aims of this study were to validate SAPS3 in COVID-19 patients, investigate the performance of SAPS3 across different waves of COVID-19 and compare the discriminative capacity of SAPS3 Box 1 and 3. Further, we wanted to investigate when persistent critical illness empirically occurs in COVID-19 in order to accurately estimate to what extent persistent critical illness contributes to ICU length of stay and mortality in COVID-19.

# Method

## Ethical considerations

This study was approved by the National Ethical Review Agency, Stockholm, Sweden (DNR: 202100352 and Ö44-2021/3.1). As only anonymized data collected as a part of routine quality control was available to the researchers, consent for participation was waived by the National Ethical Review Agency. Patients may at any time opt out of the Swedish Intensive Care Registry and have their data removed. The declaration of Helsinki and its subsequent revisions were followed, as was the STROBE statement for reporting [18]. The entire dataset is available to researchers upon request with attached ethical approval from the Swedish Intensive Care Registry, at www.icuregswe.org. Swedish law does not support sharing of individual level registry data, and thus our ethical approval from the Swedish Ethical Review Authority does not support public uploading of individual-level data as it contains sensitive patient information.

## Patient cohort

The Swedish Intensive Care Registry (SIR) is a national quality register to which all Swedish intensive care units report patients [19]. All patients with an ICU admission diagnosis of COVID-19, U07.1, according to the International Statistical Classification of Diseases and

Related Health Problems 10th edition (ICD-10), admitted between 6 March 2020 and 9 November 2021 were included. If a patient had multiple ICU admissions for COVID-19, information was collected from the first admission. No patients were excluded (S1 Fig).

## Exposure, covariates and outcome

Patient data, such as age, sex, parameters from SAPS3 and chronic health conditions, e.g. chronic heart, lung, kidney or liver disease, as well as hypertension, hyperlipidemia, diabetes mellitus, obesity, neuromuscular disease, impaired immune function and malignancy, were drawn from SIR. COVID-19 affected Sweden in three waves [5], and the date with the nadir in number of patients admitted to ICUs nationally was chosen to separate waves. Patients were followed for 30 days after ICU admission for mortality.

## Statistical analysis

SAPS3 was evaluated by calculating the area under the receiver operating characteristic (AUC) using the roccomp Stata command employing the method of DeLong et al [20]. Model goodness-of-fit was evaluated using Hosmer-Lemeshow test, the Brier score, and calibration belt [21]. SAPS3 is designed to predict hospital mortality, but as hospital mortality was not available, 30-day mortality was chosen as the outcome. The AUC for SAPS3 across waves of COVID-19 was calculated and compared using the roccomp command.

We estimated the onset of persistent critical illness by comparing after how many days of ICU stay prior health conditions predicted death better than acute physiological derangement [7] in two ways: first by comparing SAPS3 Box 1 and Box 3, and second by comparing two new custom prognostic models, one containing prior illness and one containing acute physiological derangement.

The AUC for Box 1 and Box 3 of SAPS3 was first compared in the entire cohort and then compared in patients with zero or more days of ICU stay, one or more days of ICU stay, etc. until 30 days or more of ICU stay, and plotted to ascertain after how many days Box 1 better predicted death than Box 3.

Subsequently two prognostic models, one containing prior physical illness and one containing acute physiological derangement, were created using the first wave as the development dataset and waves two and three as validation sets, similar to the method of Iwashyna et al [7]. The TRIPOD statement was followed in creating the predictive models [22]. Comorbidities were handled as binary variables, continuous variables were handled using restricted cubic splines with knots placed at the 10th, 50th and 90th percentile [23] using the xbrcspline command. Candidate predictors were chosen based on clinical relevance, and a logistic regression model with 30-day mortality as the outcome was constructed using backward elimination and Akaike Information Criterion [24]. The AUC for the prior illness model and the acute model was then calculated in patients with zero or more days of ICU stay, one or more days of ICU stay, etc. until 30 days of ICU stay and plotted to ascertain when prior physical illness better predicted death than acute physiological derangement.

The final prior illness model included age, sex, chronic lung disease, impaired immune system, chronic liver disease, diabetes, obesity and cancer. The final acute physiological derangement model contained vasoactive medication prior to ICU admission, highest body temperature, lowest systolic blood pressure, highest serum bilirubin, highest serum creatinine, highest leucocyte count, lowest thrombocyte count, lowest pH, lowest $PaO_2$, invasive mechanical ventilation prior to ICU admission, number of symptomatic days prior to ICU admission and days in hospital prior to ICU admission.

Missing data in the prior illness model was <1% for all variables, but a few candidate variables for the acute physiological derangement model had substantial proportions of missing data: $FiO_2$ 32.6%, highest bilirubin 12.1%, and mechanical ventilation prior to ICU admission 10.2%. Missing for all other variables were <10%. For the analysis comparing SAPS3 Box 1 and Box 3, 7 895 patients had all prerequisite information to be included. In the main analysis of the custom prior illness model, 2 569 patients from wave 1 had all requisite information to be included in the development dataset and 5 322 patients from wave 2 and 3 were included in the validation dataset. For the custom acute physiological derangement model, 1 950 were included in the development dataset and 4 243 in the validation dataset. Sensitivity analyses were performed using single value normal imputation as applied in the field [7–9] (yielding n = 2 589 for development and n = 5 380 for validation in both models); and complete-case analysis, only including patients with complete data in all candidate predictors for both the prior illness and acute physiological derangement models (n = 1 950 for development and n = 4 243 for validation in both models). All analyses were performed in Stata 15.1 (Stata Corp, College Station, Texas, USA).

## Results

During the study period, 7 969 patients were treated at a Swedish ICU with COVID-19 as the admission diagnosis. Patient characteristics are shown in Table 1. The median age was 63 years (IQR 52 to 71) and 70% were men. The most common prior medical conditions were hypertension (44.1%), hyperlipidemia (28.9%) and diabetes mellitus (25.4%). The median SAPS3 score at admission was 54 (IQR 48 to 61), with a median $pO_2/FiO_2$ ratio of 12.3 kPa (IQR 9.4 to 16.7). Patients scored a median of 17 points (IQR 13 to 21) on Box 1 and 10 (IQR 5 to 14) points on Box 3 of SAPS3. Median length of ICU stay was 8.1 days (IQR 2.9 to 16.9), and 4 804 (60.3%) patients were at some point treated with invasive mechanical ventilation. Of the 7 969 patients included in the study, 1 566 (19.7%) died while treated in the ICU and 1878 (23.6%) died within 30 days from ICU admission.

### SAPS3 in COVID-19

The complete SAPS3 score had AUC 0.75 (95% CI 0.74 to 0.76) for 30-day mortality (Fig 1). There was no difference in discrimination between waves: wave 1 AUC 0.73 (95% CI 0.71 to 0.75); wave 2 AUC 0.76 (95% CI 0.74 to 0.78); and wave 3 AUC 0.76 (95% CI 0.73 to 0.78), p = 0.22. The Hosmer-Lemeshow test was significant (p = 0.0038) and the Brier score was 0.15, indicating poor model fit. Calibration belt graphs with expected values on the x-axis plotted against observed values on the y-axis indicated under prediction in low-risk patients and over prediction in high-risk patients (Fig 2). Box 1 (prior physical illness) predicted 30-day mortality better than Box 3 (acute physiological derangement): AUC 0.74 (95% CI 0.72 to 0.75) vs 0.65 (95% CI 0.63 to 0.66) *P*<0.0001 in the entire cohort (Fig 1). Calibration was somewhat poorer in Box 1 (Hosmer-Lemeshow *P*<0.0001, Brier score 0.16) than Box 3 (Hosmer-Lemeshow *P* = 0.27, Brier score 0.17).

### Onset of persistent critical illness in COVID-19

SAPS3 Box 1 predicted death better than Box 3 during ICU stay (Fig 3).

Two new prognostic models were developed (OR for predictors in S1–S4 Figs). The prior illness model had AUC 0.76 (95% CI 0.74 to 0.78) and the acute model AUC 0.69 (95% CI 0.68 to 0.71) (*P*<0.0001 for difference) in the validation dataset. Prior physical illness predicted death better than acute physiological derangement throughout ICU stay, (S5 Fig) indicating that COVID-19 patients had persistent critical illness from ICU day of admission according to the prior definition. Calibration was similar: Hosmer-Lemeshow test statistically was

**Table 1. Patient characteristics.**

| | |
|---|---|
| No | 7969 |
| Age, yrs (IQR) | 63.0 (52.0–71) |
| Male sex, n (%) | 5579 (70.0) |
| ICU LOS, days (IQR) | 8.1 (2.9–16.9) |
| Symptomatic days before ICU, (IQR) | 10.0 (7.0–13) |
| Invasive mechanical ventilation treatment during ICU stay, n(%) | 4804 (60.3) |
| Renal replacement therapy during ICU stay, n(%) | 609 (7.9) |
| Steroid treatment during ICU stay, n(%) | 4633 (58.2) |
| From SAPS3: | |
| SAPS3 Score (IQR) | 54.0 (48.0–61) |
| P/F-ratio, kPa (IQR) | 12.3 (9.4–16.7) |
| PaO2, kPa (IQR) | 8.8 (7.7–10.4) |
| Heart frequency, bpm (IQR) | 92.0 (80–109) |
| Systolic blood pressure, mmHg (IQR) | 120.0 (105–140) |
| Creatinine, μmol/L (IQR) | 71.0 (57–94) |
| Leucocyte count, *10^9/L (IQR) | 9.0 (6.5–12.2) |
| Platelet count, *10^9/L (IQR) | 238.0 (181–311) |
| pH, (IQR) | 7.45 (7.40–7.48) |
| Days admitted before ICU (IQR) | 2.0 (0.0–4) |
| Vasoactive drugs before ICU admission, n(%) | 339 (4.3) |
| Unplanned admission, n(%) | 7734 (97.1) |
| Mechanical ventilation before ICU admission, n(%) | 4530 (63.3) |
| Intra-hospital location before ICU admission, n(%) | |
| Emergency department | 1943 (24.6) |
| ICU | 154 (2) |
| Intermediary care | 916 (11.6) |
| Operating theatre | 110 (1.4) |
| Ward | 4772 (60.4) |
| Prior physical illness: | |
| Hyperlipidemia, n (%) | 2302 (28.9) |
| Heart disease, n (%) | 1229 (15.4) |
| Lung disease, n (%) | 1328 (16.7) |
| Impaired immune system, n (%) | 617 (7.8) |
| Liver disease, n (%) | 75 (0.9) |
| Kidney disease, n (%) | 463 (5.8) |
| Diabetes mellitus, n (%) | 2025 (25.4) |
| Obesity, n (%) | 655 (8.2) |
| Neuromuscular disease, n (%) | 103 (1.3) |
| Hypertension, n (%) | 3515 (44.1) |
| Cancer, n (%) | 176 (2.2) |

significant for both; Brier score was 0.15 for the prior illness model and 0.17 for the acute physiological derangement model.

Sensitivity analyses showed similar results as the main analysis (S6 & S7 Figs).

## Discussion

The main finding in this nation-wide registry cohort study was that Box 1 from SAPS3, representing prior illness, predicted death better than the acute physiological derangement of Box 3

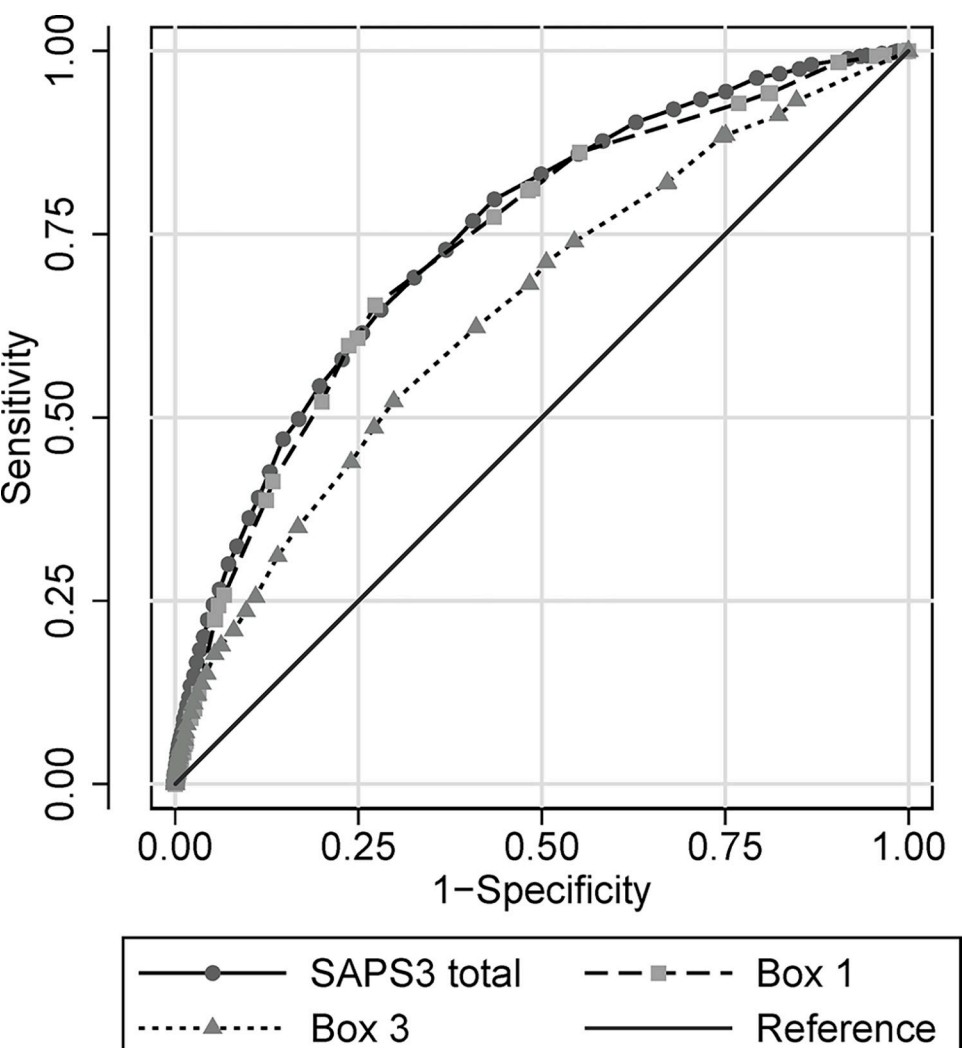

**Fig 1. Receiver operating characteristics for Simplified Acute Physiology Score 3 (SAPS3) in total, and Box 1 and Box 3 of SAPS3, in the entire cohort.**

during ICU stay for COVID-19. The whole SAPS3 score was not significantly better than Box 1 alone. The findings were reproduced using a custom prognostic model of prior physical illness that predicted death with better discrimination during ICU stay than a model containing variables describing acute physiological derangement on ICU admission. Thus, if the definition used for persistent critical illness in the literature [7] is appropriate in COVID-19 patients, they would have persistent critical illness already at ICU admission.

Earlier studies evaluating SAPS3 in COVID-19 have shown acceptable AUC values ranging from 0.69–0.83, which is consistent with our findings. Further, we replicate the rather poor calibration of SAPS3, where it under predicts mortality in low to intermediate risk patients and over predicts risk in high-risk patients [15–17], i.e. that patients with low predicted risk of death in reality have higher risk of dying than expected and vice versa in patients with high predicted risk of death. In contrast to a single-center study from Brazil, the present study found SAPS3 to have similar prognostic performance across waves of COVID-19 [25]. Despite different viral strains, different patient case mixes, pressure on ICU resources and evolving

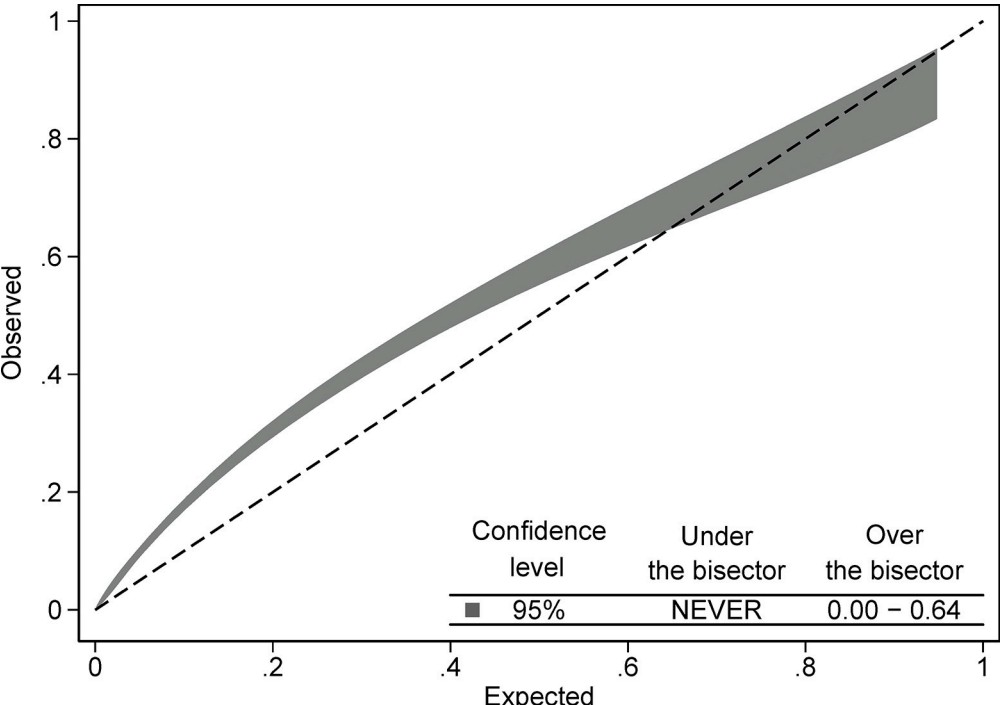

**Fig 2. Calibration belt plot of Simplified Acute Physiology Score 3 (SAPS3) in total in the entire cohort.**

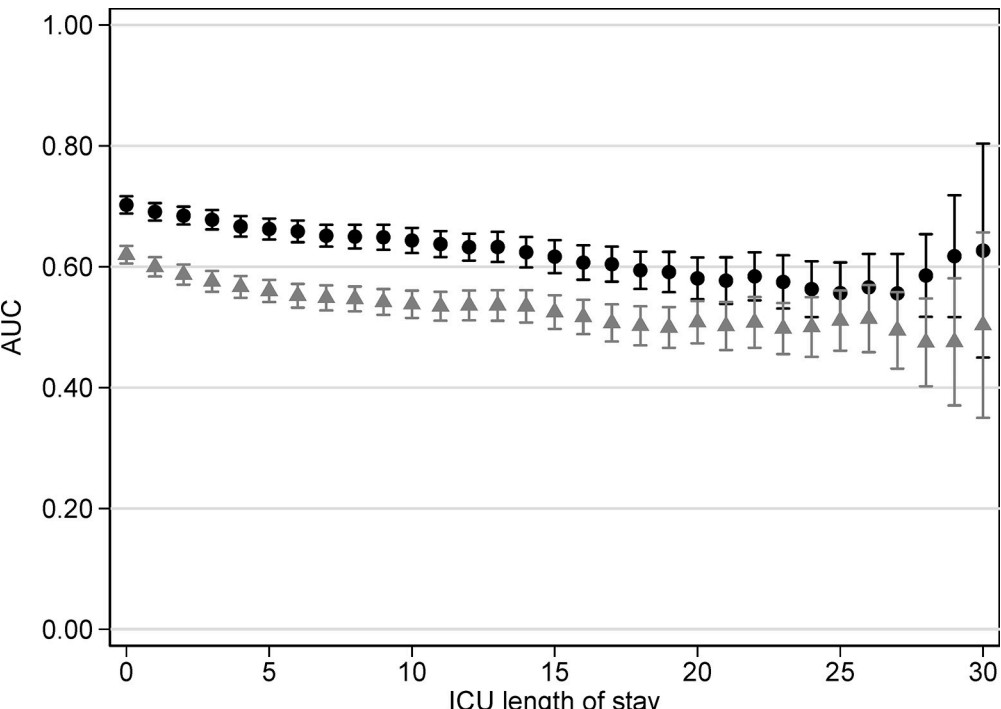

**Fig 3. Receiver operating characteristic for Box 1 and Box 3 of Simplified Acute Physiology Score 3 versus days in the ICU in the entire cohort.** Box 1 black circles, Box 3 grey triangles. AUC: area under the receiver operating characteristic.

practice of care, SAPS3 performed equally well across the waves of COVID-19. This indicates that SAPS3 may be a useful severity score in future waves of COVID-19. Further, no previous study has compared the Boxes constituting SAPS3. Interestingly, prior physical illness predicted death better than acute physiological derangement. This may be due to the pathophysiology of COVID-19, where isolated hypoxic respiratory failure was the most common cause of ICU admission [2–4], which would result in a low score in Box 3, combined with age being a strong negative prognostic factor [26].

The finding that prior physical illness predicted death better than acute physiological derangement on ICU admission during ICU stay stands in contrast to non-COVID-19 settings. This seems to be due to poor discrimination by the acute physiological derangement model: similarly to previous studies, our prior physical illness model exhibited an AUC of about 0.70, which was stable over ICU stay [7–11]. On the other hand, acute physiological derangement predicted death less well at all times, which stands in contrast to previous non-COVID-19 studies where acute models demonstrated very high AUC during the first few days followed by rapid deterioration of performance [7–11]. This difference may be due to that critically ill patients were treated at general wards for longer during the pandemic before ICU admission and thus had already begun to develop persistent critical illness; due to poor predictive capacity of acute physiological derangement in COVID-19; or due to that we are measuring the wrong predictors. Further, initial mortality was relatively rare in COVID-19 with a slow but steady worsening often over several days. The indices of severity of acute illness available for model construction in the present study were the components of SAPS3, which was designed for a general ICU population [14]. Such general risk prediction models have rather poor performance in ARDS [27, 28]. Previous prognostic models in COVID-19 have also included "conventional" vital signs, such as pulse rate, blood pressure and respiratory rate [26]. As COVID-19 most commonly presents as acute respiratory failure [2–4], future prognostic models should focus on evaluating respiratory parameters, such as respiratory rate; oxygen saturation; $pCO_2$; radiographic signs of disease severity; details of ventilatory support, such as flow and $FiO_2$ in case of high-flow oxygen, PEEP (positive end expiratory pressure) and lung compliance in case of invasive mechanical ventilation etc.; and need of adjuncts to ventilatory support such as prone positioning and muscle relaxation.

An alternative explanation for our results is that prior physical illness indeed predicts death better than acute physiological derangement in COVID-19. One definition of persistent critical illness is when the reason for continuing ICU care is more related to ongoing critical illness rather than the cause for ICU admission, where a cascade of complications has rendered the admitting diagnosis and its associated physiological derangement less relevant [6]. This may be highly relevant in the case of COVID-19, where the slow progression and long duration of disease as well as the slow recovery of respiratory function leads to prolonged ICU stay with an increased risk of complications, exacerbation of underlying conditions and declining physical function that patients with prior physical illness and advanced age are poorly equipped to compensate for. Alternatively, our inability to reproduce previous findings in the non-COVID-19 setting may indicate that this method of empirically finding the onset of persistent critical illness is not universally applicable.

The main strengths of the present study is the national registry-based approach, yielding a high number of patients with minimal loss to follow-up and strong external validity. It is to our knowledge one of the largest studies evaluating SAPS3 in COVID-19, the first to compare SAPS3 across all waves of COVID-19 and compare SAPS3 Boxes as well as the first to try to empirically find the onset of persistent critical illness in COVID-19.

Limitations include possible under-reporting of comorbid conditions, most notably obesity, in the register. Unfortunately there was substantial amounts of missing information

concerning acute physiological derangement, but sensitivity analyses using both single value normal imputation and complete-case analysis showed results similar to the main analysis. Comparison with previous studies may be hampered by the extraordinary circumstances during the COVID-19 pandemic, with its unprecedented shortage of ICU beds.

## Conclusions

In conclusion, in this nationwide ICU registry study prior physical illness predicted death better than severity of acute illness during ICU stay. Either prior physical illnesses are indeed more important for the prognosis in COVID-19 than acute physiological derangement, or conventionally collected measures of acute physiological derangement poorly capture disease severity in COVID-19. To further investigate this question future studies of prognostic models for COVID-19 may use novel parameters that better quantifies the severity of acute respiratory disease.

## Supporting information

**S1 Fig. Cohort flowchart.**
(PDF)

**S2 Fig. Unadjusted odds ratios (OR) and 95% confidence intervals (CI) for all categorical predictors.**
(PDF)

**S3 Fig. Adjusted odds ratios (OR) and 95% confidence intervals (CI) for all categorical predictors.** Predictors adjusted for all other variables in the respective model: sex, chronic lung disease, impaired immune system, chronic liver disease, diabetes, obesity and cancer adjusted for each other and age. Vasoactive medication prior to ICU admission and mechanical ventilation before ICU adjusted for each other and highest body temperature, lowest systolic blood pressure, highest bilirubin, highest creatinine, highest leucocyte count, lowest thrombocyte count, lowest pH, lowest $PaO_2$, number of symptomatic days prior to ICU admission and days in hospital prior to ICU admission.
(PDF)

**S4 Fig. Unadjusted odds ratios (OR) and 95% confidence intervals (CI) for all continuous predictors.** Presented using restricted cubic splines with knots at the 10[th], 50[th] and 90[th] percentile.
(PDF)

**S5 Fig. Receiver operating characteristic for the new prognostic models comprising prior illness and acute physiological derangement versus days in the ICU in the validation dataset.** Prior illness: black circles, acute physiological derangement: grey triangles. AUC: area under the receiver operating characteristic.
(PDF)

**S6 Fig. Receiver operating characteristic for the new prognostic models comprising prior illness and acute physiological derangement versus days in the ICU in the validation dataset.** Only includes complete cases with all information necessary to be included in both the prior and acute models. Prior illness: black circles, acute physiological derangement: grey triangles. AUC: area under the receiver operating characteristic.
(PDF)

**S7 Fig. Receiver operating characteristic for the new prognostic models comprising prior illness and acute physiological derangement versus days in the ICU in the validation dataset.** Missing data replaced with single imputation of a normal value. Prior illness: black circles, acute physiological derangement: grey triangles. AUC: area under the receiver operating characteristic.
(PDF)

## Acknowledgments

We acknowledge all participating ICUs in the Swedish Intensive Care Registry for their participation and hard work to contribute data.

## Author Contributions

**Conceptualization:** Miklos Lipcsey.

**Data curation:** Karl Stattin.

**Formal analysis:** Karl Stattin.

**Funding acquisition:** Michael Hultström, Robert Frithiof.

**Investigation:** Karl Stattin, Michael Hultström, Robert Frithiof, Miklos Lipcsey, Rafael Kawati.

**Methodology:** Karl Stattin, Michael Hultström, Robert Frithiof, Miklos Lipcsey, Rafael Kawati.

**Supervision:** Michael Hultström, Miklos Lipcsey, Rafael Kawati.

**Visualization:** Karl Stattin.

**Writing – original draft:** Karl Stattin.

**Writing – review & editing:** Karl Stattin, Michael Hultström, Robert Frithiof, Miklos Lipcsey, Rafael Kawati.

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
