## [Decision Letter · Decision Letter 0]

21 Mar 2023

PONE-D-23-04974Prior physical illness predicts death better than acute physiological derangement on intensive care unit admission in COVID-19: a Swedish registry studyPLOS ONE

Dear Dr. Stattin,

Thank you for submitting your manuscript to PLOS ONE. After careful consideration, we feel that it has merit but does not fully meet PLOS ONE’s publication criteria as it currently stands. Therefore, we invite you to submit a revised version of the manuscript that addresses the points raised during the review process.

We look forward to receiving your revised manuscript.

Kind regards,

Zivanai Cuthbert Chapanduka, MBChB (M.D)

Academic Editor

PLOS ONE

Journal Requirements:

Additional Editor Comments:

Dear Karl Stattin

The first round of review is complete.

Kindly respond to the reviewer queries comprehensively.

Please note that Plos1 allows 2 review rounds.

Thank you

Reviewers' comments:

Reviewer's Responses to Questions

**Comments to the Author**

1. Is the manuscript technically sound, and do the data support the conclusions?

Reviewer #1: Partly

Reviewer #2: Partly

2. Has the statistical analysis been performed appropriately and rigorously? 

Reviewer #1: Yes

Reviewer #2: Yes

3. Have the authors made all data underlying the findings in their manuscript fully available?

Reviewer #1: Yes

Reviewer #2: Yes

4. Is the manuscript presented in an intelligible fashion and written in standard English?

Reviewer #1: Yes

Reviewer #2: No

5. Review Comments to the Author

Reviewer #1: STATISTICAL ANALYSIS

LINE138 - 139 For the prior illness model, 2 569 patients had all requisite information to be included in the development dataset and 5 322 individuals were included in the validation

Clarify the difference between cases referred to as patients and those referred as individuals?

The total cohort was 7 969, state how many were excluded.

LINE 141 - 143 Sensitivity analyses were performed using single value normal imputation as applied in the field (7-9)(yielding n=2 589 for development and n=5 380 for validation in both models);

What does n represent (patients or individuals)?

RESULTS

LINE 157 - 158. During ICU stay 1 566 (19.7%) patients died, whereas 1 878 (23.6%) died within 30 days of ICU admission. What is the difference between the two groups?

LINE 160. Mechanical ventilation before ICU admission, n (%) 4530 (63.3%).

How long were these patients on mechanical ventilation before ICU admission?

Did these patients not meet the criteria for ICU admission?

A study in USA found a significant association between the availability of hospital resources- particularly ICU beds - and patient mortality during the early weeks of COVID-19 pandemic: https://doi.org/10.12788/jhm.3539

Did you consider delayed ICU admissions as one of the limitation when comparing Box 1 (prior physical illness) and Box 3 (acute physiological derangement)

Likewise, these patients (n = 4530; 63.3%) could have been admitted to ICU with acute physiological derangements, supporting the idea that COVID-19 patients would have persistent critical illness already at ICU admission.

LINE 187 - 188. The prior illness model included age, sex, chronic lung disease, impaired immune system, chronic liver disease, diabetes, obesity and cancer.

Hypertension (44.1%) and hyperlipidaemia (28.9%) were the most common prior medical conditions in the study, why are the two excluded in this model?

LINE 194 – 197. Prior physical illness predicted death better than acute physiological derangement at all times during ICU stay, (Supplementary Figure 4) indicating that COVID-19 patients had persistent critical illness from ICU day of admission according to the prior definition. Do you perhaps have a data of patients with more than one comorbidities such as hypertension/hyperlipidaemia/diabetes mellitus?

In view of the following major Limitations in your manuscript:

1.Obesity: under-reported

2.Acute physiological derangements: substantial amounts of missing information

3.ICU admission: unprecedented shortage of ICU beds

It is well-studied that mortality of Covid-19 patients is associated prior physical illness (supported by Gupta et al – LINE 365 and Van der Merwewith very important to

Were patients without above data used to finalise the conclusion of “Prior physical illness predicts death better than acute physiological derangement”? if yes, indicate what percentage of patients had unavailability of both obesity data and acute physiological derangement data.

Reviewer #2: Please see attachment.

1.In the Background section of the abstract, the authors mention that this “…has not been studied in COVID-19” (line 27) but mentions in line 214 in the Discussion that “Earlier studies evaluating SAPS3 in COVID-19 have shown acceptable AUC values ranging from 0.69-0.83, which is consistent with our findings.” Are the authors suggesting that these referenced studies are different? Clarification is required in terms of how this study is different from others or has not been studied in COVID-19.

2.In the Methods section of the abstract (line 33) the authors use AUC ROC as the abbreviation for “area under receiver operating characteristic” but later in the text only AUC is used as an abbreviation. Suggest to use only one abbreviation if referring to one concept, alternatively new abbreviations need to be clarified/explained.

3.Please clarify the result section of abstract (lines 39 - 41): do these two AUC values refer to the Box1 or Box3 components of SAPS3 and/or the trained results on wave-1 patients and tested results on wave-2/3 patients. What are the two variables compared with each other?

4.Lines 85 – 89: Please provide a heading for the ethics statement under the Method section e.g. “Ethical considerations”

5.The method section requires refinement and sufficient detail for better understanding.

a.The total sample size (entire cohort) – patients admitted to ICU between the given dates (?n=7969) – The authors only mention this in the Results section and not in the Method section

i.The authors are unclear (lines 138-146) on how n=2569 for the prior illness model in the development dataset and n=5322 in the validation dataset was selected. Please clarify the exclusion and inclusion criteria for each dataset as well as for each model. (The authors should provide the eligibility criteria, and the sources and methods of selection of participants in the method section)

ii.Following the sensitivity analyses and exclusion of individuals, the development data set was n=1950 and the validation data set was n=4243 for both models

iii.A flow diagram/chart may help with clarification or better presentation of each stage of the number of individuals/participants and how the authors got to this final cohort that is being used for the statistical analysis

iv.The authors do not mention that wave-1 patients are being used for the development data set and wave-2/3 patients being used for the validation data set in the Method section.

1.The following is mentioned in line 40 of the abstract only: “…custom logistic models trained on wave-1 patients (n=?) and tested in wave-2 and -3 (n=?) patients.” Does this refer to development and validation data sets respectively?

v.Report numbers of individuals at each stage of the study and reasons for non-participation/exclusion at each stage of the study

b.The authors mention that two prognostic models were compared: 1. Prior physical illness (? chronic model/?Box1) and 2. Acute physiological derangement (?acute model/Box3)

i.Later, in the text these models are alternatively referred to as acute and chronic models (lines 198 and 199). The authors are advised to clarify this in the method section and/or only use one term to describe a model to avoid confusion later in the text.

SUGGEST: revise the Method section

6.Lines 154-155 “Patients scored a median of 17 points (IQR 13 to 21) on Box 1 (of which median 9 (IQR 5 to 13) from age) and 10 (IQR 5 to 14) points on Box 3 of SAPS3.”: This sentence seems to have some missing information and values are not available in Table 1 to find out. Please also explain what these values mean and what is their significance.

a.Please use alternative brackets “[]” within another set of brackets.

7.Line 157: Does “mechanical ventilation” refer to all patients that were mechanically ventilated but not necessarily intubated?

8.Line 157-158: “During ICU stay 1 566 (19.7%) patients died whereas 1 878 (23.6%) died within 30 days of ICU admission”

a.I am not sure what the authors are trying to say with these numbers.

b.Please explain how the total number of patients who died during the ICU stay (n=1566) is less than the number of patients who died within 30 days of ICU admission (n=1878). Does “ICU stay” refer to a longer period than the 30 days that are mentioned in line 105?

c.Line 105: “Patients were followed for 30 days after ICU admission for mortality.” And because “…hospital mortality was not available, 30-day mortality was chosen as the outcome.” (line 111)

d.Are these values (percentages) proportions of the total cohort, even if they were not included in the statistical analysis?

9.Line 166: Please provide AUC values and their precision (95% confidence intervals) for the statement that there was no difference in discrimination between waves.

10.Line 168-169: The calibration belt graph shown in Figure 2 does not clearly illustrate the finding

a.Please clarify this with clear Y-axis and X-axis titles and values in the text.

11.Line 171: Please indicate that this is referring to Figure 1, if this is so.

12.Line 180: SAPS3 Box 1 predicted death better than Box 3 at all times during ICU stay

a.Are there statistically significant differences between Box1 and Box3 after day 20 when the 95% confidence intervals overlap to make this statement “at all times” ?

b.Please provide statistical justification for this statement.

13.Are the values shown in Figure 1 and Figure 3 results from the development dataset or the validation dataset? Please clarify if this is indeed based on the development dataset.

14.Lines 186-192: Consider this information in the Method section rather than under “Results”.

a.Describe how the two models were developed and the selection of participants in the method section.

15.Line 195: Is there a statistically significant difference between the two models beyond day 20? Please justify or clarify this statement.

a.Please provide numerical results in addition to the graphs for the AUC of ICU length of stay at 30 days.

b.This is similar to the point made in no 12.

16.Line 219: Please consider the following article and reconsider the statement: “No previous study has compared…”

Ana Paula Pires Lázaro, Polianna Lemos Moura Moreira Albuquerque, Gdayllon Cavalcante Meneses, et al. Critically ill COVID-19 patients in northeast Brazil: mortality predictors during the first and second waves including SAPS 3, Transactions of The Royal Society of Tropical Medicine and Hygiene, Volume 116, Issue 11, November 2022, Pages 1054–1062, https://doi.org/10.1093/trstmh/trac046

17.Line 224: Repetition of lines 205-206 or is this a different statement based on a different data set? Please clarify.

This is an interesting article with good results that should be considered for publication after the major and minor changes suggested.

6. PLOS authors have the option to publish the peer review history of their article (what does this mean?). If published, this will include your full peer review and any attached files.

Reviewer #1: No

Reviewer #2: No

---

## [Author Response · Author response to Decision Letter 0]

5 Apr 2023

Response to Reviewers

We would like to thank the Reviewers for their time and expertise and the Editor for the opportunity to improve our manuscript. We have made extensive changes to the text in order to address the comments and believe it has greatly improved the manuscript. We hope you find the changes satisfactory. Please see attached file with changes highlighted.

The authors, through Karl Stattin

We apologize. The manuscript text and file names have been changed to adhere to PLOS ONE style requirements.

The need for consent was waived by the Swedish Ethical Review Authority, as only anonymized data collected as a part of routine quality control was available to researchers. Patients may opt out of the Swedish Intensive Care Registry at any time and have their data removed.

The entire dataset is available to researchers upon request with attached ethical approval from the Swedish Intensive Care Registry, at www.icuregswe.org. Our ethical approval from the Swedish Ethical Review Authority does not support public uploading of individual-level data as it contains sensitive patient information.

Captions have been added and the manuscript changed accordingly.

Reviewer #1: STATISTICAL ANALYSIS

LINE138 - 139 For the prior illness model, 2 569 patients had all requisite information to be included in the development dataset and 5 322 individuals were included in the validation

Clarify the difference between cases referred to as patients and those referred as individuals?

We apologise for the confusion. There is no difference, all included individuals are patients treated at an ICU. The manuscript has been edited for clarity, using only “patients” throughout.

The total cohort was 7 969, state how many were excluded.

No patients were excluded. The Methods section has been edited to clarify this. 

LINE 141 - 143 Sensitivity analyses were performed using single value normal imputation as applied in the field (7-9)(yielding n=2 589 for development and n=5 380 for validation in both models);

What does n represent (patients or individuals)?

“n” represents patients. The manuscript has been edited to only refer to “patients”.

RESULTS

LINE 157 - 158. During ICU stay 1 566 (19.7%) patients died, whereas 1 878 (23.6%) died within 30 days of ICU admission. What is the difference between the two groups?

ICU stay refers to the time the patient is treated in the ICU. When the patient is discharged to a ward, the patient may no longer be included in this statistic. Conversely, 30-day mortality refers to 30 calendar days after ICU admission, irrespective of where the patient is. The sentence has been rephrased for clarity.

LINE 160. Mechanical ventilation before ICU admission, n (%) 4530 (63.3%).

How long were these patients on mechanical ventilation before ICU admission?

Did these patients not meet the criteria for ICU admission?

Mechanical ventilation before ICU admission can be invasive or noninvasive mechanical ventilation. In Sweden during this period in time, general wards and emergency departments did not have the resources to provide invasive mechanical ventilation for any extended period of time. As such, any invasive mechanical ventilation prior to ICU admission was initiation of invasive mechanical ventilation just before or during transport to the ICU.

During the COVID-19 pandemic, Swedish hospitals acquired the resources to provide noninvasive mechanical ventilation on most wards, mainly High-Flow Nasal Oxygen therapy (HFNO). Unfortunately, the duration of such therapy is not documented in the Swedish Intensive Care Registry and was therefore not available to us. These patients may have met habitual ICU admission criteria during non-pandemic circumstances, but we have previously shown in this cohort1 that during pandemic surges, ICU beds were reserved for sicker patients requiring more intensive respiratory support modalities. This did however not affect mortality.

1. Stattin et al. Strain on the ICU resources and patient outcomes in the COVID-19 pandemic. European Journal of Anaesthesiology. 2023;40(1):13-20

A study in USA found a significant association between the availability of hospital resources- particularly ICU beds - and patient mortality during the early weeks of COVID-19 pandemic: https://doi.org/10.12788/jhm.3539

Did you consider delayed ICU admissions as one of the limitation when comparing Box 1 (prior physical illness) and Box 3 (acute physiological derangement)

Likewise, these patients (n = 4530; 63.3%) could have been admitted to ICU with acute physiological derangements, supporting the idea that COVID-19 patients would have persistent critical illness already at ICU admission.

In this cohort, we have previously shown that ICU strain (ICU beds available, ICU bed occupancy, etc) was not associated with increased mortality in Sweden during this period in time.1 It is possible that the association was different in the USA during the first weeks of the pandemic. Nevertheless, this is a very interesting point: perhaps the patients were critically ill already on the ward and came to the ICU later in the disease process. This line of reasoning has been added to the discussion.

1. Stattin et al. Strain on the ICU resources and patient outcomes in the COVID-19 pandemic. European Journal of Anaesthesiology. 2023;40(1):13-20

LINE 187 - 188. The prior illness model included age, sex, chronic lung disease, impaired immune system, chronic liver disease, diabetes, obesity and cancer.

Hypertension (44.1%) and hyperlipidaemia (28.9%) were the most common prior medical conditions in the study, why are the two excluded in this model?

Hypertension and hyperlipidaemia were indeed the most common medical conditions, but they did not reach statistical significance when we constructed our custom prognostic models. We designed the custom prognostic models using backwards elimination in a logistic regression model and the Akaike Information Criterion1. In brief, this method entails first constructing a large statistical model where all variables that may possibly affect patient mortality are included and the model performance is tested. Next, the variable with the weakest association with mortality is removed, and the performance of the new reduced model is tested. This step is then repeated again and again, removing the weakest remaining variable each time, until the model performance is no longer improved. 

The other variables (age, sex, chronic lung disease, etc) were stronger predictors of death than hypertension and hyperlipidaemia, and were thus included in the model. 

1. Royston P. Prognosis and prognostic research: Developing a prognostic model. BMJ 2009;338 

LINE 194 – 197. Prior physical illness predicted death better than acute physiological derangement at all times during ICU stay, (Supplementary Figure 4) indicating that COVID-19 patients had persistent critical illness from ICU day of admission according to the prior definition. Do you perhaps have a data of patients with more than one comorbidities such as hypertension/hyperlipidaemia/diabetes mellitus?

Many patients have more than one comorbidity. The prior physical illness model predicts mortality based on the “comorbidity pattern” of each patient, i.e. if a patient has only one comorbidity, it predicts mortality based on that illness. If a patient has two (or more) conditions, it adds the risks of all conditions to calculate the total risk. We apologise if this was not clear in the manuscript.

In view of the following major Limitations in your manuscript:

1.Obesity: under-reported

2.Acute physiological derangements: substantial amounts of missing information

3.ICU admission: unprecedented shortage of ICU beds

It is well-studied that mortality of Covid-19 patients is associated prior physical illness (supported by Gupta et al – LINE 365 and Van der Merwewith very important to

Were patients without above data used to finalise the conclusion of “Prior physical illness predicts death better than acute physiological derangement”? if yes, indicate what percentage of patients had unavailability of both obesity data and acute physiological derangement data.

We agree that missing data may hamper the interpretation of studies. Further, we believe that obesity may be under-reported, i.e. patients that were obese are not reported as obese. This means that obesity does not have missing values, but the Registry may misclassify patients as non-obese (this would likely not change the results however, as obesity is a risk factor for COVID-19 and misclassifying obese individuals as non-obese would diminish the difference between compared groups). 

To test the robustness of our main analysis we therefore performed two sensitivity analyses: a single-value normal imputation analysis (where missing data was replaced with a normal value, including 2 598 patients in the development dataset and 5 380 patients in the validation dataset) and a complete-case analysis (where only patients with complete both prior illness data and acute physiological data were included, including 1 950 patients in the development dataset and 4 243 patients in the validation dataset). These analyses take into account missingness in obesity and acute physiological derangement. Both these analyses showed results similar to the main analysis, supporting our conclusion.

Reviewer 2:

1. In the Background section of the abstract, the authors mention that this “…has not been studied in COVID-19” (line 27) but mentions in line 214 in the Discussion that “Earlier studies evaluating SAPS3 in COVID-19 have shown acceptable AUC values ranging from 0.69-0.83, which is consistent with our findings.” Are the authors suggesting that these referenced studies are different? Clarification is required in terms of how this study is different from others or has not been studied in COVID-19.

The abstract refers to that the onset of persistent critical illness has not been empirically defined in COVID-19, but this was not clearly expressed. The Abstract has been edited for clarity.

2. In the Methods section of the abstract (line 33) the authors use AUC ROC as the abbreviation for “area under receiver operating characteristic” but later in the text only AUC is used as an abbreviation. Suggest to use only one abbreviation if referring to one concept, alternatively new abbreviations need to be clarified/explained. 

We are indeed referring to one concept, and should therefore use only one abbreviation. The manuscript has been edited for consistency, only using “AUC”.

3. Please clarify the result section of abstract (lines 39 - 41): do these two AUC values refer to the Box1 or Box3 components of SAPS3 and/or the trained results on wave-1 patients and tested results on wave-2/3 patients. What are the two variables compared with each other? 

The first pair of AUC values compared indeed represent SAPS3 Box 1 vs Box 3. The latter pair of AUC values compare the custom prior illness model vs the custom acute physiological derangement model. The Abstract was poorly worded, and has been rephrased for clarity.

4. Lines 85 – 89: Please provide a heading for the ethics statement under the Method section e.g. “Ethical considerations” 

A heading reading “Ethical considerations” has been added.

5. The method section requires refinement and sufficient detail for better understanding. 

a. The total sample size (entire cohort) – patients admitted to ICU between the given dates (?n=7969) – The authors only mention this in the Results section and not in the Method section

i. The authors are unclear (lines 138-146) on how n=2569 for the prior illness model in the development dataset and n=5322 in the validation dataset was selected. Please clarify the exclusion and inclusion criteria for each dataset as well as for each model. (The authors should provide the eligibility criteria, and the sources and methods of selection of participants in the method section)

According to the STROBE statement1 Item 13(a), number of patients should be reported under Results, and not Methods.

The inclusion and exclusion criteria are outlined in the Methods section under the heading “Patient cohort”, lines 96-101. The division into development and validation dataset is described under “Statistical methods” line 128-139. The Methods section has been edited for clarity, explaining how patients were selected for the models.

1. Vandenbroucke JP, von Elm E, Altman DG, Gotzsche PC, Mulrow CD, Pocock SJ, et al. Strengthening the Reporting of Observational Studies in Epidemiology (STROBE): explanation and elaboration. Epidemiology. 2007;18(6):805-35.

ii. Following the sensitivity analyses and exclusion of individuals, the development data set was n=1950 and the validation data set was n=4243 for both models 

These numbers refer to the complete-case sensitivity analysis, where only individuals with all information in both the prior illness model and the acute physiological derangement model were included. 

iii. A flow diagram/chart may help with clarification or better presentation of each stage of the number of individuals/participants and how the authors got to this final cohort that is being used for the statistical analysis

Excellent suggestion. A flowchart has been added to the supplementary material.

iv. The authors do not mention that wave-1 patients are being used for the development data set and wave-2/3 patients being used for the validation data set in the Method section. 

Important point. This is mentioned line 128-131 in the Methods section. 

1. The following is mentioned in line 40 of the abstract only: “…custom logistic models trained on wave-1 patients (n=?) and tested in wave-2 and -3 (n=?) patients.” Does this refer to development and validation data sets respectively? 

Yes. This wording has been removed from the sentence in order to make the paragraph easier to understand, in accordance with comment 3. 

v. Report numbers of individuals at each stage of the study and reasons for non-participation/exclusion at each stage of the study

This has been added to the supplementary flowchart, and the Methods section edited to underline this.

b. The authors mention that two prognostic models were compared: 1. Prior physical illness (? chronic model/?Box1) and 2. Acute physiological derangement (?acute model/Box3)

i. Later, in the text these models are alternatively referred to as acute and chronic models (lines 198 and 199). The authors are advised to clarify this in the method section and/or only use one term to describe a model to avoid confusion later in the text. 

We agree that the use of different wordings may cause confusion. The manuscript has been edited for consistency, using “acute physiological derangement” and “prior illness” throughout.

SUGGEST: revise the Method section

6. Lines 154-155 “Patients scored a median of 17 points (IQR 13 to 21) on Box 1 (of which median 9 (IQR 5 to 13) from age) and 10 (IQR 5 to 14) points on Box 3 of SAPS3.”: This sentence seems to have some missing information and values are not available in Table 1 to find out. Please also explain what these values mean and what is their significance. 

a. Please use alternative brackets “[]” within another set of brackets.

SAPS3 Box 1 (prior illness) includes points for age, but not Box 3 (acute physiological derangement). As age is such an important risk factor for death in COVID-19, we thought it would be interesting for readers to know how many points were scored for age. We now realise that this wording may be a source of confusion and adds little to the manuscript, and we have therefore removed it.

Table 1 lists the actual values of patient parameters on ICU admission rather than the SAPS3 points scored for these parameters, as we thought this would be more informative to readers. 

7. Line 157: Does “mechanical ventilation” refer to all patients that were mechanically ventilated but not necessarily intubated? 

Mechanical ventilation in this instance refers to invasive mechanical ventilation. The wording was unclear, and the manuscript has been edited to specify “invasive mechanical ventilation”. 

8. Line 157-158: “During ICU stay 1 566 (19.7%) patients died whereas 1 878 (23.6%) died within 30 days of ICU admission”

a. I am not sure what the authors are trying to say with these numbers.

Of 7 969 patients included in the study, 1 566 (19.7%) patients died while treated in the ICU, whereas 1878 (23.6%) patients died within 30 days from the day they were admitted to the ICU. The sentence has been edited for clarity. 

b. Please explain how the total number of patients who died during the ICU stay (n=1566) is less than the number of patients who died within 30 days of ICU admission (n=1878). Does “ICU stay” refer to a longer period than the 30 days that are mentioned in line 105?

ICU stay refers to the time the patient is being treated in the ICU. If the patient is discharged to a ward or home, they are no longer included in this statistic. ICU stay is often shorter than 30 days.

Conversely, 30-day mortality refers to calendar days from the day of ICU admission, irrespective of where the patient is treated. The manuscript has been edited to clarify this.

c. Line 105: “Patients were followed for 30 days after ICU admission for mortality.” And because “…hospital mortality was not available, 30-day mortality was chosen as the outcome.” (line 111)

SAPS3 was originally designed to predict hospital mortality. Unfortunately, this information was not available to us. To approximate hospital mortality, we chose to compute 30-day mortality instead. We realise this may not have been apparent, and has edited the manuscript for clarity.

d. Are these values (percentages) proportions of the total cohort, even if they were not included in the statistical analysis?

These are percentages of the total cohort. The total cohort was included in the single value normal imputation analysis. We have edited the Results to clarify the denominator. 

9. Line 166: Please provide AUC values and their precision (95% confidence intervals) for the statement that there was no difference in discrimination between waves. 

The AUC values and 95% confidence intervals have been added.

10. Line 168-169: The calibration belt graph shown in Figure 2 does not clearly illustrate the finding

a. Please clarify this with clear Y-axis and X-axis titles and values in the text. 

The plot, its axes and interpretation has been clarified in the manuscript.

11. Line 171: Please indicate that this is referring to Figure 1, if this is so. 

Thank you. It does, and a reference has been added.

12. Line 180: SAPS3 Box 1 predicted death better than Box 3 at all times during ICU stay

a. Are there statistically significant differences between Box1 and Box3 after day 20 when the 95% confidence intervals overlap to make this statement “at all times” ?

b. Please provide statistical justification for this statement. 

The difference is indeed not statistically significant the last days of the plot, but we believe this is due to low statistical power (most patents that die do so earlier in the disease, leaving fewer in the analysis in the last days) and not due to actual changes in performance of the prognostic models. As such we do not believe it alters the interpretation of the results, but the manuscript has been edited to reflect this uncertainty.

13. Are the values shown in Figure 1 and Figure 3 results from the development dataset or the validation dataset? Please clarify if this is indeed based on the development dataset. 

Figure 1 and 3 concerns SAPS3, which is not derived from this dataset. It thus reflects the entire cohort. The manuscript has been edited to clarify this.

14. Lines 186-192: Consider this information in the Method section rather than under “Results”.

a. Describe how the two models were developed and the selection of participants in the method section. 

The section has been moved. Model development is described in line 128-139.

15. Line 195: Is there a statistically significant difference between the two models beyond day 20? Please justify or clarify this statement. 

a. Please provide numerical results in addition to the graphs for the AUC of ICU length of stay at 30 days. 

b. This is similar to the point made in no 12. 

See response to comment 12. The last days of the graph has lower statistical power due to few patients, and although the difference does not reach statistical significance, we do not believe it alters the overall interpretation of the results. The manuscript has been altered to reflect this uncertainty.

16. Line 219: Please consider the following article and reconsider the statement: “No previous study has compared…”

Ana Paula Pires Lázaro, Polianna Lemos Moura Moreira Albuquerque, Gdayllon Cavalcante Meneses, et al. Critically ill COVID-19 patients in northeast Brazil: mortality predictors during the first and second waves including SAPS 3, Transactions of The Royal Society of Tropical Medicine and Hygiene, Volume 116, Issue 11, November 2022, Pages 1054–1062, https://doi.org/10.1093/trstmh/trac046

Thank you for that reference, it escaped our literature review. It has been added to the reference list and the manuscript changed accordingly.

17. Line 224: Repetition of lines 205-206 or is this a different statement based on a different data set? Please clarify.

Line 205-206 is a general introduction to the Discussion section of the manuscript and line 224 is a deeper discussion of the results.

This is an interesting article with good results that should be considered for publication after the major and minor changes suggested.

Thank you for your time and expertise. We feel your comments have greatly improved our manuscript. We hope you agree and that all comments have been sufficiently addressed.

---

## [Decision Letter · Decision Letter 1]

9 May 2023

PONE-D-23-04974R1Prior physical illness predicts death better than acute physiological derangement on intensive care unit admission in COVID-19: a Swedish registry studyPLOS ONE

Dear Dr.Stattin

Thank you for submitting your revised manuscript to PLOS ONE. After careful consideration, we feel that it has merit but does not fully meet PLOS ONE’s publication criteria as it currently stands. Therefore, we invite you to submit a revised version of the manuscript that addresses the points raised during the review process.

We look forward to receiving your revised manuscript.

Kind regards,

Zivanai Cuthbert Chapanduka, MBChB (M.D)

Academic Editor

PLOS ONE

Journal Requirements:

Additional Editor Comments:

Thanks you for the revisions which the reviewer has seen and commented on. The review is almost complete. Kindly attend to the reviewer's requirements CAREFULLY, ensuring that no new problems arises. Please note that Plos1 limits the number of reviews to 2 and that this is the final review round.

All the best

Reviewers' comments:

Reviewer's Responses to Questions

**Comments to the Author**

1. If the authors have adequately addressed your comments raised in a previous round of review and you feel that this manuscript is now acceptable for publication, you may indicate that here to bypass the “Comments to the Author” section, enter your conflict of interest statement in the “Confidential to Editor” section, and submit your "Accept" recommendation.

Reviewer #1: All comments have been addressed

Reviewer #2: All comments have been addressed

2. Is the manuscript technically sound, and do the data support the conclusions?

Reviewer #1: Yes

Reviewer #2: Yes

3. Has the statistical analysis been performed appropriately and rigorously? 

Reviewer #1: Yes

Reviewer #2: Yes

4. Have the authors made all data underlying the findings in their manuscript fully available?

Reviewer #1: Yes

Reviewer #2: Yes

5. Is the manuscript presented in an intelligible fashion and written in standard English?

Reviewer #1: Yes

Reviewer #2: Yes

6. Review Comments to the Author

Reviewer #1: Future submission:

Please ensure that your manuscript meets PLOS ONE's style requirements.

Please specify where the minimal data set underlying the results described in your manuscript can be found.

Reviewer #2: Lines 136-137: The authors mentioned in response to comment number 7 of Reviewer 2, that mechanical ventilation only refers to invasive mechanical ventilation and that the manuscript has been edited to specify this. However, I am unable to note this in the revised manuscript.

The revised manuscript shows significant improvements and no further major revisions are proposed. This original research addresses a relevant and important question that is clearly answered by the results and discussion, with conclusions supported by the data.

7. PLOS authors have the option to publish the peer review history of their article (what does this mean?). If published, this will include your full peer review and any attached files.

Reviewer #1: **Yes: **Dr Ernest Musekwa

Reviewer #2: No

---

## [Author Response · Author response to Decision Letter 1]

11 May 2023

Dear Editor and Reviewers,

Thank you for your time and expertise helping us improve our article. We have addressed all comments, we hope satisfactorily. Please see attached manuscript with changes highlighted.

Sincerely,

The authors, through Karl Stattin 

Additional Editor Comments:

Thanks you for the revisions which the reviewer has seen and commented on. The review is almost complete. Kindly attend to the reviewer's requirements CAREFULLY, ensuring that no new problems arises. Please note that Plos1 limits the number of reviews to 2 and that this is the final review round.

All the best

Thank you for the opportunity to review our manuscript. References have been updated in order to adhere to the PLOS ONE style guidelines. None of the cited references have been redacted.

Reviewer #1: Future submission:

Please ensure that your manuscript meets PLOS ONE's style requirements.

Please specify where the minimal data set underlying the results described in your manuscript can be found.

Thank you, we will be more careful to follow style requirements. Availability of the minimal dataset has been added to the paragraph “Ethical considerations” line 87-91: The entire dataset is available to researchers upon request with appropriate ethical approval from the Swedish Intensive Care Registry, at www.icuregswe.org. Swedish law does not support sharing of individual level registry data, and thus our ethical approval from the Swedish Ethical Review Authority does not support public uploading of individual-level data as it contains sensitive patient information.

Reviewer #2: Lines 136-137: The authors mentioned in response to comment number 7 of Reviewer 2, that mechanical ventilation only refers to invasive mechanical ventilation and that the manuscript has been edited to specify this. However, I am unable to note this in the revised manuscript.

The revised manuscript shows significant improvements and no further major revisions are proposed. This original research addresses a relevant and important question that is clearly answered by the results and discussion, with conclusions supported by the data.

Thank you for your thorough revision. We apologise for the omission. Line 140 has been edited to specify invasive mechanical

---

## [Editor Report · Decision Letter 2]

23 May 2023

PONE-D-23-04974R2Prior physical illness predicts death better than acute physiological derangement on intensive care unit admission in COVID-19: a Swedish registry studyPLOS ONE

Dear Dr. Stattin

Thank you for submitting your manuscript to PLOS ONE. After careful consideration, we feel that it has merit but does not fully meet PLOS ONE’s publication criteria as it currently stands. Therefore, we invite you to submit a revised version of the manuscript that addresses the points raised during the review process.

Please submit your revised manuscript by Jul 07 2023 11:59PM If you will need more time than this to complete your revisions, please reply to this message or contact the journal office at plosone@plos.org. Please include the following items when submitting your revised manuscript:A rebuttal letter that responds to each point raised by the academic editor and reviewer(s). You should upload this letter as a separate file labeled 'Response to Reviewers'.A marked-up copy of your manuscript that highlights changes made to the original version. You should upload this as a separate file labeled 'Revised Manuscript with Track Changes'.An unmarked version of your revised paper without tracked changes. You should upload this as a separate file labeled 'Manuscript'.If applicable, we recommend that you deposit your laboratory protocols in protocols.io to enhance the reproducibility of your results. Protocols.io assigns your protocol its own identifier (DOI) so that it can be cited independently in the future. For instructions see: https://journals.plos.org/plosone/s/submission-guidelines#loc-laboratory-protocols. Additionally, PLOS ONE offers an option for publishing peer-reviewed Lab Protocol articles, which describe protocols hosted on protocols.io. Read more information on sharing protocols at https://plos.org/protocols?utm_medium=editorial-email&utm_source=authorletters&utm_campaign=protocols.

We look forward to receiving your revised manuscript.

Kind regards,

Zivanai Cuthbert Chapanduka, MBChB (M.D)

Academic Editor

PLOS ONE
---

## [Author Response · Author response to Decision Letter 2]

7 Sep 2023

Dear Editor and Reviewers,

Thank you for your time and expertise helping us improve our article. We have addressed all comments, we hope satisfactorily. Please see attached manuscript with changes highlighted.

Sincerely,

The authors, through Karl Stattin 

Editorial comments

Additional Editor Comments:

Thanks you for the revisions which the reviewer has seen and commented on. The review is almost complete. Kindly attend to the reviewer's requirements CAREFULLY, ensuring that no new problems arises. Please note that Plos1 limits the number of reviews to 2 and that this is the final review round.

Thank you for the opportunity to review our manuscript. References have been updated in order to adhere to the PLOS ONE style guidelines. None of the cited references have been redacted.

Reviewer #1: Future submission:

Please ensure that your manuscript meets PLOS ONE's style requirements.

Please specify where the minimal data set underlying the results described in your manuscript can be found.

Thank you, we will be more careful to follow style requirements. Availability of the minimal dataset has been added to the paragraph “Ethical considerations” line 87-91: The entire dataset is available to researchers upon request with appropriate ethical approval from the Swedish Intensive Care Registry, at www.icuregswe.org. Swedish law does not support sharing of individual level registry data, and thus our ethical approval from the Swedish Ethical Review Authority does not support public uploading of individual-level data as it contains sensitive patient information.

Reviewer #2: Lines 136-137: The authors mentioned in response to comment number 7 of Reviewer 2, that mechanical ventilation only refers to invasive mechanical ventilation and that the manuscript has been edited to specify this. However, I am unable to note this in the revised manuscript.

The revised manuscript shows significant improvements and no further major revisions are proposed. This original research addresses a relevant and important question that is clearly answered by the results and discussion, with conclusions supported by the data.

Thank you for your thorough revision. We apologise for the omission. Line 140 has been edited to specify invasive mechanical ventilation.

Sincerely,

Karl Stattin, MD, PhD, EDAIC

---

## [Editor Report · Decision Letter 3]

14 Sep 2023

Prior physical illness predicts death better than acute physiological derangement on intensive care unit admission in COVID-19: a Swedish registry study

PONE-D-23-04974R3

Dear Dr. Statin

We are pleased to inform you that your manuscript has been judged scientifically suitable for publication and will be formally accepted for publication once it meets all outstanding technical requirements.

Kind regards,

Zivanai Cuthbert Chapanduka, MBChB (M.D)

Academic Editor

PLOS ONE
---

## [Editor Report · Acceptance letter]

19 Sep 2023

PONE-D-23-04974R3 

Prior physical illness predicts death better than acute physiological derangement on intensive care unit admission in COVID-19: a Swedish registry study 

Dear Dr. Stattin:

I'm pleased to inform you that your manuscript has been deemed suitable for publication in PLOS ONE. Congratulations! Your manuscript is now with our production department. 

Kind regards, 

on behalf of

Professor Zivanai Cuthbert Chapanduka 

Academic Editor

PLOS ONE